# Equivariant Asynchronous Diffusion: An Adaptive Denoising Schedule for Accelerated Molecular Conformation Generation

## Abstract

Recent 3D molecular generation methods primarily use asynchronous auto-regressive or synchronous diffusion models. While auto-regressive models build molecules sequentially, they're limited by a short horizon and a discrepancy between training and inference. Conversely, synchronous diffusion models denoise all atoms at once, offering a molecule-level horizon but failing to capture the causal relationships inherent in hierarchical molecular structures. We introduce Equivariant Asynchronous Diffusion (EAD) to overcome these limitations. EAD is a novel diffusion model that combines the strengths of both approaches: it uses an asynchronous denoising schedule to better capture molecular hierarchy while maintaining a molecule-level horizon. Since these relationships are often complex, we propose a dynamic scheduling mechanism to adaptively determine the denoising timestep. Experimental results show that EAD achieves state-of-the-art performance in 3D molecular generation. The code is anonymously released in https://anonymous.4open.science/r/EAD-3F4E

## 1 Introduction

Diffusion models have achieved remarkable success in modeling complex data distributions, particularly in domains such as images (Ho et al., 2020) and language (Lovelace et al., 2023). More recently, this paradigm has gained substantial traction in molecular structure generation, enabling key applications such as *de novo* molecular discovery (Hajduk & Greer, 2007) and property-guided design (Kang et al., 2006). Among the many proposed approaches, equivariant diffusion models (EDMs) (Hoogeboom et al., 2022) have emerged as a leading variant, distinguished by their ability to denoise 3D coordinates through SE(3)-equivariant graph neural networks (GNNs).

Prior to the rise of diffusion models, auto-regressive (AR) models (Gebauer et al., 2019; Luo & Ji, 2022; Daigavane et al., 2024) appeared to offer a solution by iteratively predicting the next atom or fragment until a "STOP" symbol was generated. These approaches typically establish a hierarchical structure within the molecule and perform asynchronous generation, allowing key structures (e.g., molecular scaffolds) to emerge preferentially. However, they face two main limitations: (i) a lack of a global horizon, preventing the generation of molecules under desirable conditions, and (ii) inconsistencies between training and inference that can lead to issues such as significant error accumulation in large molecular generation. In contrast, full-molecule diffusion models (Hoogeboom et al., 2022; Huang et al., 2024) overcome these challenges by denoising all atoms synchronously, enabling global condition guidance and the generation of larger molecules. Nevertheless, these models overlook the hierarchical nature of molecular structures: small uncertainties in key components (e.g., scaffolds) can introduce significant uncertainties in modified regions (e.g., functional groups), potentially causing spatial inconsistencies such as incorrect atomic bonds or angles.

Autoregressive and diffusion models exhibit complementary strengths, naturally prompting the question: **Can they be combined to enable more flexible and rational molecular generation?** Several approaches have explored this idea in domains such as language sequences and videos, where a Markov causal chain is integrated within the diffusion process, making the denoising of subsequent tokens (or frames) dependent on those previously denoised (Wu et al., 2023; Chen et al., 2024; Song et al., 2025). However, training these methods is challenging due to the vast number of

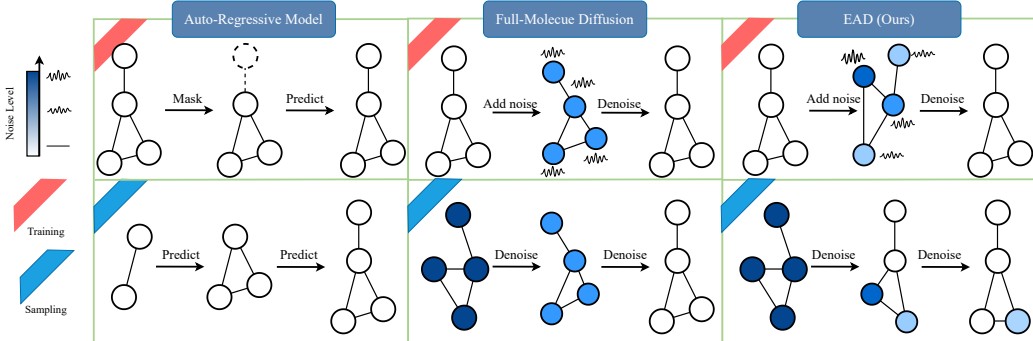

Figure 1: **Generation Processes Overview.** Left: Autoregressive methods generate atoms sequentially, with each new atom's generation conditioned on the previously generated, noise-free atoms. Middle: Full-molecule diffusion models denoise all atoms simultaneously, iteratively refining a sample of noisy atoms until they are all noise-free. Right: Our proposed EAD model combines the strengths of both approaches by using an asynchronous denoising process. This allows atoms to become noise-free sequentially, where the denoising direction for each atom can benefit from the information of atoms that have already reached a lower noise state. Details of their difference can be found in Section 3.2.

potential combinations in asynchronous timestep schedules. More importantly, this concept does not directly extend to molecular generation, as molecular data is represented as a graph, where causal relationships are implicit and significantly more complex.

In this paper, we introduce the Equivariant Asynchronous Diffusion (EAD) model for 3D molecular structure generation, which incorporates a stable asynchronous noise training paradigm and a dynamic denoising strategy. During both training and inference, different atoms can be assigned different noise levels, allowing some atoms to be denoised earlier while still being guided by global conditions. To ensure stable training, we independently sample noise levels for each atom, while imposing a range constraint to prevent the model from learning implausible noise combinations during sampling. Additionally, we propose a dynamic denoising timestep schedule based on historical denoising steps to prioritize atoms for denoising during sampling. This approach enables the diffusion model to learn hierarchical relationships while effectively minimizing cumulative errors. Figure 1 illustrates the differences between EAD and traditional methods. Notably, EAD is a general diffusion method that is independent of model architecture. Our experiments demonstrate that EAD outperforms the synchronous denoising method EDM—using the same architecture and number of training iterations—across all molecular generation metrics, particularly in molecular-level properties, with an 8% increase in molecular stability and a 3% improvement in validity. Furthermore, EAD achieves competitive performance compared to other state-of-the-art generation methods.

**Contributions** (i) EAD, a novel 3D molecular generation method that combines the strengths of both autoregressive and diffusion models; (ii) A stable asynchronous diffusion training strategy, coupled with a dynamic denoising timestep schedule that adapts based on historical denoising; and (iii) Empirical validation demonstrating EAD's potential for generating complete, valid molecules. Specifically, we show that traditional full-molecule diffusion models are special cases of EAD, and that the EAD framework can be integrated into various diffusion architectures.

## 2  PRELIMINARIES

In this section, we provide a brief overview of 3D molecular generation methods. Additionally, we introduce the notations commonly used in the following sections. The symbol $[\cdot;\cdot]$ denotes concatenation, while $q(\cdot)$ and $p(\cdot)$ represent the transition distributions. The terms $\alpha_t$ and $\sigma_t$ refer to the noise schedule in the diffusion and generation processes. We use the term "clean" to refer to samples that are either noise-free or contain only a minimal amount of noise.

## 2.1 MOLECULAR AUTOREGRESSIVE MODELS

The autoregressive model is a sequence generation model that predicts the next token based on historical information. The distribution of the $i$-th token is expressed as:

$$\mathbf{x}_i \sim p_\theta(\mathbf{x}_{j<i}). \tag{1}$$

Sequence autoregression typically formulates a sequential prediction process as $\mathbf{x}_i = \phi(\mathbf{z}_{i-1})$, where $\mathbf{z}_{i-1}$ represents the hidden state of the $i-1$-th token, and $\phi(\cdot)$ denotes a prediction model. However, this approach cannot be directly applied to molecular generation due to the uncertain interdependencies within the molecular graph structure. To address this challenge, (Gebauer et al., 2019) proposed an autoregressive generation method based on "focus atoms". The focus atom serves as the chosen origin for that step of the generation process, and its index is predicted by an additional model $g(\cdot)$:

$$j = \arg\max_{j<i} g(\mathbf{z}_j). \tag{2}$$

Training $g(\cdot)$ involves transforming the molecular graph into a sequence of subgraphs and identifying the target atom within each subgraph. The atomic attributes (atomic type and 3D coordinates) of the recent atom can then be predicted as $\mathbf{x}_i = \phi(\mathbf{z}_j)$. Note that atomic 3D coordinates lie in a continuous and unbounded space, there are two common methods for predicting them: (i) using the invariant features of multiple atoms to estimate coordinates (Gebauer et al., 2019; Luo & Ji, 2022), (ii) using an E(3)-equivariant GNN (Daigavane et al., 2024; Liao & Smidt, 2023) to directly predict the relative coordinates. Despite of graph generation, existing molecular autoregressive algorithms convert molecular graphs into subgraph sequences through algorithms such as depth-first search or nearest neighbor search. However, these methods fail to effectively represent the implicit hierarchical structure of molecules. In this work, we allow the model to adaptively identify the atoms that should be prioritized for denoising.

## 2.2 FULL-MOLECULE DIFFUSION MODELS

Diffusion model is based on a process of gradually adding noise to data and then learning to denoise to reconstruct the original data. Equivariant diffusion model (EDM) first apply this technology on 3D molecular generation (Hoogeboom et al., 2022). Given a molecular data $\mathbf{x} = [\mathbf{p}; \mathbf{h}]$ represented by atomic coordinates $\mathbf{p}$ and atomic types $\mathbf{h}$ on all the atoms, the noise process with diffusion step $t \in \{0, ..., T\}$ is defined as

$$q(\mathbf{z}_t|\mathbf{x}) = \mathcal{N}(\mathbf{z}_t|\alpha_t\mathbf{x}, \sigma_t^2\mathbf{I}), \tag{3}$$

where $\mathcal{N}$ is a product of normal distribution $\mathcal{N}^p$ and $\mathcal{N}^h$. $\mathbf{z}_t = [\mathbf{z}_t^\mathbf{p}; \mathbf{z}_t^\mathbf{h}]$ denote the noised latent variable. In general, $\alpha_t$ is modelled by a function that smoothly transitions from $\alpha_0 \approx 1$ towards $\alpha_T \approx 0$, while $\sigma_t$ behaves in the opposite way. Together, they control how much true distribution and random noise are contained in the latent variable. Note that equation 3 is a full-molecule diffusion where all the atoms share the same $t$, $\alpha_t$ and $\sigma_t$. Furthermore, EDM applies a common strategy where $\alpha_t = \sqrt{1-\sigma^2}$. This diffusion process is Markov and can be equivalently written with transition distributions as:

$$q(\mathbf{z}_t|\mathbf{z}_s) = \mathcal{N}(\mathbf{z}_t|\alpha_{t|s}\mathbf{z}_s, \ \sigma_{t|s}^2\mathbf{I}), \tag{4}$$

for any $t > s$ with $\alpha_{t|s} = \alpha_t/\alpha_s$ and $\sigma_{t|s}^2 = \sigma_t^2 - \alpha_{t|s}^2\sigma_s^2$. Hence, the entire noising process is then written as:

$$q(\mathbf{z}_0, \mathbf{z}_1, \ldots, \mathbf{z}_T|\mathbf{x}) = q(\mathbf{z}_0|\mathbf{x})\prod_{t=1}^{T} q(\mathbf{z}_t|\mathbf{z}_{t-1}). \tag{5}$$

The posterior of the transitions conditioned on $\mathbf{x}$ gives the inverse of the noising process. It is also normal and given by:

$$q(\mathbf{z}_s|\mathbf{x}, \mathbf{z}_t) = \mathcal{N}(\mathbf{z}_s|\mu_{t\to s}(\mathbf{x}, \mathbf{z}_t), \sigma_{t\to s}^2\mathbf{I}), \tag{6}$$

where the definitions for $\mu_{t\to s}(\mathbf{x}, \mathbf{z}_t)$ and $\sigma_{t\to s}$ can be analytically obtained as

$$\mu_{t\to s}(\mathbf{x}, \mathbf{z}_t) = \frac{\alpha_{t|s}\sigma_s^2}{\sigma_t^2}\mathbf{z}_t + \frac{\alpha_s\sigma_{t|s}^2}{\sigma_t^2}\mathbf{x}, \quad \sigma_{t\to s} = \frac{\sigma_{t|s}\sigma_s}{\sigma_t}$$

In equation 6, the distribution of the latent variable $\mathbf{z}_s$ can be computed by $\mu_{t\to s}(\mathbf{x}, \mathbf{z}_t)$. However, since $\mathbf{x}$ is unknown during generation, it is replaced by a dynamic function $\hat{\mathbf{x}} = \phi(\mathbf{z}_t, t)$, where $\phi(\cdot)$ is a GNN. The denoising distribution is then expressed as:

$$p(\mathbf{z}_s|\mathbf{z}_t) = \mathcal{N}(\mathbf{z}_s|\mu_{t\to s}(\hat{\mathbf{x}}, \mathbf{z}_t), \sigma_{t\to s}^2\mathbf{I}). \tag{7}$$

**Algorithm 1** Asynchronous Diffusion Training

1: **loop**
2:     Sample a molecule $([\mathbf{x}_1; \mathbf{h}_1], ..., [\mathbf{x}_M; \mathbf{h}_M])$ from dataset $\mathcal{D}$. The asynchronous interval is $C$.
3:     Sample a baseline noise level $t^*$ from $\mathcal{U}(0, ..., T)$.
4:     **for** $i = 1, ..., M$ **do**
5:         $t_i = \text{Clamp}(t_i^c + t^*)$
6:         Sample $t_i^c$ from $[-M, M]$.
7:         Sample noise $\boldsymbol{\epsilon}_i \sim \mathcal{N}(\mathbf{0}, \mathbf{I})$
8:         Extract $\alpha_{t_i}$ and $\sigma_{t_i}$ from schedule.
9:         $\mathbf{z}_i = \alpha_{t_i}[\mathbf{x}_i; \mathbf{h}_i] + \sigma_{t_i}\boldsymbol{\epsilon}_i$
10:    **end for**
11:    Concatenate atomic representation $\mathbf{z}_i$ to $\mathbf{z}$.
12:    Compute $\boldsymbol{\epsilon}^\theta = \phi(\mathbf{z}, [t_i; ...; t_M])$
13:    Minimize $\|[\boldsymbol{\epsilon}_1; ...; \boldsymbol{\epsilon}_M] - [\boldsymbol{\epsilon}_1^\theta; ...; \boldsymbol{\epsilon}_M^\theta]\|^2$
14: **end loop**

**Algorithm 2** Asynchronous Sampling

1: **Input:** Model $\phi(\cdot)$, denoising history $\mathbf{h} = None$.
2: **Initialize:** Maximum iteration $K$, denoising iteration $k = K$, $\mathbf{z}^K \sim \mathcal{N}(\mathbf{0}, \mathbf{I})$, atom-wise timstep $\mathbf{t} = [T + 1; ...; T + 1]$, asynchronous ratio $\lambda$.
3: **while** $\sum_{i=1}^M t_M > 0$ or $k > 0$ **do**
4:     **if** $k > (\lambda K)$ **then**
5:         $\mathbf{t} := \mathbf{t} - 1$
6:     **else**
7:         $\mathbf{v} = \text{Compare}(\mathbf{h}, \mathbf{h}^*)$, and $\mathbf{t} := \mathbf{t} - \mathbf{v}$
8:     **end if**
9:     $\mathbf{t} := \text{Clamp}(\mathbf{t})$, and $\mathbf{s} = \mathbf{t} - 1$
10:    Sample $\boldsymbol{\epsilon} \sim \mathcal{N}(\mathbf{0}, \mathbf{I})$
11:    $\mathbf{z}^{k-1} = \frac{1}{\alpha_{\mathbf{t}|\mathbf{s}}}\mathbf{z}^k - \frac{\sigma_{\mathbf{t}|\mathbf{s}}^2}{\alpha_{\mathbf{t}|\mathbf{s}}\sigma_{\mathbf{t}}}\phi(\mathbf{z}^k, \mathbf{t}) + \sigma_{\mathbf{t}\to\mathbf{s}}\boldsymbol{\epsilon}$
12:    Compute $\mathbf{h}^*$ by $g(\mathbf{z}^{k-1}, \mathbf{z}^k)$ and update $\mathbf{h}$
13:    $k := k - 1$
14: **end while**
15: Sample $\mathbf{x}, \mathbf{h} \sim p(\mathbf{x}, \mathbf{h}|\mathbf{z}^0)$
16: **Return** $\mathbf{x}, \mathbf{h}$.

The neural network is optimized using $\mathcal{L}(\mathbf{x}, \phi(\mathbf{z}_t, t))$ based on the true data distribution. EDM adopts a similar approach, but instead of directly estimating the posterior, it predicts the noise:

$$\boldsymbol{\epsilon}^{\mathbf{P}}, \boldsymbol{\epsilon}^{\mathbf{h}} = \phi(\mathbf{z}_t, t) - [\mathbf{z}_t^{\mathbf{P}}; \mathbf{0}], \tag{8}$$

where the attribute noise $\boldsymbol{\epsilon}^{\mathbf{h}}$ is predicted directly, and the positional noise is computed by subtracting the input coordinates. This ensures that the output resides within a zero-centered gravity subspace. After optimization, EDM employs a standard denoising procedure with $s = t - 1$, where all atoms follow the same denoising schedule. As illustrated in Figure 1(2), EDM as a full-molecule diffusion method does not account for the atom order; rather, all atoms are optimized simultaneously into noise-free states by the end of the process. Further details on 3D molecular diffusion, including SO(3)-equivariance, dynamic models, and optimization conditions, are provided in the Appendix B.

# 3 EQUIVARIANT ASYNCHRONOUS DIFFUSION MODEL

In this section, we first make a review of hierarchical structure in molecule to detail our motivation. Then, we introduction the formulation of our EAD and clarify its properties.

## 3.1 REVISITING THE HIERARCHICAL STRUCTURE IN MOLECULE

The structure of organic molecules is inherently hierarchical, comprised of functional groups and structural scaffolds that form a complex, interconnected whole (Lim et al., 2020). This hierarchical nature also exists at the atomic level, where atoms like carbon and oxygen play a different, more foundational role than hydrogen atoms. Some motif-based graph methods (Zhang et al., 2021; 2023) have leveraged these hierarchies to create more effective molecular representations. In generation tasks, hierarchical structures used to construct causal chains to determine the priority of atom generation (Daigavane et al., 2024). This is in contrast to domains like language or video, where the explicit sequential order of tokens or frames introduces a clear causal relationship (Chen et al., 2024; Song et al., 2025). However, the relationships within a molecular hierarchy are implicit and represented by a graph, which lacks a fixed order. Consequently, imposing a rigid, chain-like denoising order would misrepresent the multifaceted nature of molecular formation, which is defined by multiple potential paths rather than a single linear one. We empirically confirm this claim in Section 5.4. In this work, we define a more flexible approach to finding the order of molecular generation. Based on the diffusion framework, we can dynamically adjust priorities during the denoising process.

## 3.2 THE DIFFUSION PROCESS

The process of generating a 3D molecule requires simultaneously modeling two types of data: discrete atomic types ($\mathbf{h}$) and continuous coordinates ($\mathbf{p}$). The latent variables, which include noise, are defined as $\mathbf{z}_i^t = [\mathbf{z}_i^{t,(\mathbf{p})}; \mathbf{z}_i^{t,(\mathbf{h})}]$, where $i$ denotes the atom index and $t \in [0, T]$ denotes the noise level or noise timestep. A common approach is to use E(n)-Equivariant Graph Neural Networks (EGNNs) (Satorras et al., 2021c) to model the dynamics of this diffusion process, a method that has shown success in 3D tasks (Hoogeboom et al., 2022). Our model adopts this dynamic architecture, whose full-molecule diffusion is given by:

$$\boldsymbol{\epsilon}_i^\theta = \phi(\{\mathbf{z}_j^t | j \in 1, ..., M\}, t). \tag{9}$$

This formulation predicts the noise $\boldsymbol{\epsilon}_i^\theta$ for atom $i$ based on the state of all atoms in the molecule at the same noise level $t$. However, this holistic dependence, which is a hallmark of full-sequence diffusion methods (Ho et al., 2022; Janner et al., 2022; Li et al., 2022), can create a significant challenge: it may lead to inconsistencies and physically implausible relationships between neighboring atoms in the generated structure.

**Independent Noise Level**    Unsynchronized denoising is a critical component of the EAD model. To achieve this, we apply diverse noise levels to the different atoms during the diffusion process, simliar to (Chen et al., 2024). Specifically, for the $i$-th atom, the corresponding latent variables $\mathbf{z}_{i,t_i}$ are defined as:

$$q(\mathbf{z}_i^{t_i}|\mathbf{x}_i) = \mathcal{N}(\mathbf{z}_i^{t_i}|\alpha_{t_i}\mathbf{x}_i, \sigma_{t_i}^2\mathbf{I}), \tag{10}$$

where $t_i$ is independently sampled from $\mathcal{U}(0, ..., T)$. This setup allows the dynamic model to capture all possible combinations of atomic noise levels. However, training this model is intuitively more challenging than training full-molecule diffusion methods, as the number of possible noise combinations is significantly greater than that of synchronous diffusion.

We present a constrained independent sampling method designed to optimize training efficiency, as detailed in Algorithm 1. The procedure is initiated by sampling a global noise level $t^*$ from a uniform distribution $\mathcal{U}(0, ..., T)$. Subsequently, a narrow asynchronous interval $[-C, C]$, with $C \ll T$, is usually defined as the maximal molecular size in training set. For each atom, a local noise offset $t_i^c$ is independently sampled from this interval. The final noise level for atom $i$ is then determined by $t_i = t_i^c + t^*$. All noise levels are clamped to the range $[0, T]$ to ensure adherence to the valid parameter space. In the following, we use the vector $\mathbf{t} = [t_0, ...t_M]$ to denote the noise level of molecule, and the dynamic function becomes $\boldsymbol{\epsilon}^\theta = \phi(\mathbf{z}^\mathbf{t}, \mathbf{t})$ containing state of all atoms. This strategy significantly reduces the combinatorial complexity of noise configurations from $\mathcal{O}(T^M)$ to $\mathcal{O}((2C)^M)$, with the primary objective of guiding the model to learn a more constrained and probable subset of asynchronous noise levels. We further impose explicit constraints on the sampling process to ensure consistency. As our experimental findings in Section 5 demonstrate, this approach yields a significant improvement in sampling performance over the base EDM model, without requiring additional training epochs.

**Optimization Objective**    During training, we noise data distribution according to equation 10 and use the dynamic model to learn denoising. The loss function for this process can be expressed as:

$$\mathcal{L}_{Diff} = \mathop{\mathbb{E}}_{\mathbf{t}, \mathbf{x}, \boldsymbol{\epsilon}} \frac{1}{M} \sum\nolimits_{i=1}^{M} \Big[ \|\boldsymbol{\epsilon}_i - \phi(\mathbf{z}^\mathbf{t}, \mathbf{t})_i\|^2 \Big], \tag{11}$$

where $\boldsymbol{\epsilon}_i \sim \mathcal{N}(\mathbf{0}, \sigma_t^2\mathbf{I})$. Additionally, we incorporate common techniques such as the likelihood term $\mathcal{L}_0 = \log p(\mathbf{x}|\mathbf{z}_0)$ and the distribution distance of the final latent variable, $\mathcal{L}_{Final} = -KL(q(\mathbf{z}_T|\mathbf{x}), p(\mathbf{z}_\mathbf{T}))$ (Hoogeboom et al., 2022).

Compared to full-molecule diffusion, **the training of EAD represents a more universal learning paradigm. It learns across diverse noise levels, which enables the use of various denoising timestep schedules during sampling without the need for model retraining.** In our experiments in Section 5.4, we evaluated synchronous schedule, manual asynchronous schedule, and adaptive asynchronous schedules, all of which performed well. Importantly, these results were achieved with a single training run.

**Variable-Size Generation**   Full-molecule diffusion methods typically require molecules to be padded with "dummy atoms" during training. These atoms are usually not involved in the model's computations but serve to define the molecule's size during the sampling phase. While (Hoogeboom et al., 2022) proposed a method that pre-samples the number of dummy atoms to generate different molecular sizes, this approach is not sufficiently flexible, particularly in scenarios where the level of target molecular size is unknown.

Our EDM model introduces a new method for variable-size generation by actively including dummy atoms in the computational process, allowing the model to learn their denoising dynamics. Given that dummy atoms function as "STOP" symbols that should appear only after valid atoms, we increase the likelihood of them being assigned higher noise levels during training process. Specifically, we modify the independent noise sampling for dummy atoms, giving it a 50% probability of being drawn from $[0, C]$ instead of the standard $[-C, C]$. This design empowers our model to automatically predict the molecular size and ensures that it prioritizes the generation of valid molecular structures during the sampling process.

## 3.3 SAMPLING

We outline the sampling process for EAD in Algorithm 2. This method continuously updates a denoising history state $\mathbf{h}$ that guides the independent denoising trajectory of each atom. We partition the total denoising process in into two stages. The first stage begins with a standard Gaussian noise representation $\mathbf{z}^K$, setting the noise level of all atoms to $T$. We first iteratively perform synchronous denoising for a certain step to obtain a preliminary state for the subsequent stage. In the second stage, each atom leverages its historical state $\mathbf{h}_i$ to determine whether to advance its current noise level. This historical state encodes the feature dynamics of the $i$-th atom during denoising and can be conceptualized as the velocity of its corresponding atomic distribution. We define the velocity of $i$-th atom as

$$\mathbf{h}^* = g(\mathbf{z}_i^{k-1}, \mathbf{z}_i^k) = \|\mathbf{z}_i^{k-1} - \mathbf{z}_i^k\|^2. \tag{12}$$

The denoising process corresponds to a progressive transition from a noise distribution to a clean data distribution (Song et al., 2020). A key property of this convergence is that the velocity of the distribution should asymptotically approach zero. Based on this principle, we flag atoms whose velocity does not exhibit a monotonic decrease. For these "stalled" atoms, we pause their noise timestep, performing multiple denoising steps at the same noise level until a stable downward velocity trend is re-established. Compare($\cdot$) in Algorithm 2 is used to assess the convergence of the velocity, which is defined as

$$\text{Compare}(\mathbf{h}_i, \mathbf{h}_i^*) = \begin{cases} 1 & \text{if } \mathbf{h}_i^* \leq \min(\mathbf{h}_i) \\ 0 & \text{otherwise.} \end{cases} \tag{13}$$

Here $\mathbf{h}$ retains the most recent portion of the velocity through a window of size $w$. This adaptive mechanism allows for the automatic generation of a molecular hierarchy during the denoising process, where the most structurally resolved components are fully denoised first ($t_i = 0$).

**Boundary**   During training, we define the asynchronous interval as $[-C, C]$. To prevent combinations that exceed this interval during sampling, we clamp noise levels of all atoms within a molecule to the interval $[\min(t_1, ..., t_M), \min(t_1, ..., t_M) + 2C]$. This clamp forcibly advance noise levels of atoms whose velocity remains unstable for a long time. Additionally, for atoms that are already clean ($t_i = 0$), we freeze their features to prevent them from deviating from the correct data distribution.

## 4 RELATED WORK

Most methods for 3D molecular structure generation fall into one of two broad categories: autoregressive and end-to-end models.

**Autoregressive Methods**   G-SchNet (Gebauer et al., 2019; 2022) and G-SphereNet (Luo & Ji, 2022) were early successes in autoregressive molecular structure generation. G-SchNet uses the SchNet (Schütt et al., 2017) framework with rotationally invariant features for message passing, generating node embeddings. Atoms in the current fragment then "vote" on the next atom's placement within a 3D grid by specifying radial distances from a central "focus node." However, this approach requires

at least three atoms for triangulation due to the rotationally invariant features, necessitating additional tokens to break symmetry. G-SphereNet similarly performs triangulation using normalizing flows once three atoms are present. SYMPHONY (Daigavane et al., 2024) addresses this limitation with higher-degree E(3)-equivariant features, which describe atomic states more completely and thus eliminate the need for triangulation.

**Diffusion-based Methods**   We focus on the diffusion end-to-end models, which is closely related to ours. Diffusion models are designed to learning to reverse a process that progressively adds noise to molecular structures until a simple, tractable distribution is reached. The model then learns to denoise, enabling the generation of new molecules through iterative refinement. GeoDiff (Xu et al., 2022) generates 3D molecular conformations in Euclidean space. TorsionDiff (Jing et al., 2022) applies the diffusion process to torsion angles while keeping other degrees of freedom fixed. EDM (Hoogeboom et al., 2022) denoises both continuous atomic coordinates and categorical atom types. GEOLDM (Xu et al., 2023) extends this by performing diffusion in the latent space of an equivariant Variational Auto-Encoder (VAE). Researchers have also explored equivariant neural networks within the diffusion framework for tasks like molecular linker design (Igashov et al., 2024), leveraging E(3) equivariance to ensure consistency with molecular symmetries (e.g., rotations and translations) and improve physical plausibility.

## 5 EXPERIMENTS

In this section, we conduct experiments to evaluate the performance of EAD across diverse generation tasks, datasets, and sampling strategies. We compare our model with a variety of diffusion-based methods that operate in 3D space, including EDM (Hoogeboom et al., 2022), EDM's non-equivariant variant GDM, EDM-Bridge (Wu et al., 2022), GeoLDM (Xu et al., 2023), and UniGEM (Feng et al., 2025), which is the most architecturally similar to our approach. For a broader comparison, we also include other prominent generative methods: the flow-based E-NF (Garcia Satorras et al., 2021) and the autoregressive G-SchNet (Gebauer et al., 2022).

Table 1: Comparison of generation performance on the QM9 dataset, including atom stability, molecule stability, validity, and validity*uniqueness. Higher values indicate better performance. Gray shading denotes the base model used for EAD.

| Model | Atom sta(%) | Mol sta(%) | Valid(%) | V*U(%) | Sec./sample |
|---|---|---|---|---|---|
| E-NF | 85.0 | 4.9 | 40.2 | 39.4 | - |
| G-Schnet | 95.7 | 68.1 | 85.5 | 80.3 | - |
| EDM | 98.7 ±0.1% | 82.0 ±0.4% | 91.9 ±0.5% | 90.7 ±0.6% | 0.20 |
| GDM | 97.6 | 71.6 | 90.4 | 89.5 | - |
| EDM-Bridge | 98.8 | 84.6 | 92.0 | 90.7 | - |
| GeoLDM | 98.9 | 89.4 | 93.8 | 92.7 | - |
| UniGEM | **99.0** | 89.8 | 95.0 | **93.2** | - |
| EAD (Ours) | **99.0** ±0.1% | **90.3** ±0.1% | **95.1** ±0.2% | **93.2** ±0.2% | 0.21 |
| Data | 99.0 | 95.2 | 97.7 | 97.7 | - |

### 5.1 RESULTS ON QM9

**Dataset and Configurations**   We evaluate EAD's generative capabilities on the QM9 dataset (Ramakrishnan et al., 2014), a standard 3D molecular dataset containing 130k small molecules with up to nine heavy atoms, including molecular properties and atomic coordinates. EAD is trained to generate molecules with 3D coordinates and atom types (H, C, N, O, F). We use the train/validation/test splits from (Anderson et al., 2019), consisting of 100k, 18k, and 13k samples, respectively. The diffusion process uses $T = 1000$ steps, the asynchronous ratio is set to $0.8$ and the sampling window is $w = 2$. Further configuration follow the EDM (Hoogeboom et al., 2022) whose details are in Appendix D.1.

**Metrics**   We evaluate our model by sampling 10,000 molecules and measuring four key metrics: atom stability, molecule stability, validity, and uniqueness. First, inter-atomic distances are used to predict bond types. Atom stability is then defined as the percentage of atoms with correct valency,

while molecule stability is the fraction of generated molecules where every atom is stable. Validity and uniqueness are determined by first converting the 3D molecular structures to SMILES format using RDKit. Uniqueness is then calculated as the ratio of unique molecules among all valid, non-duplicate samples.

**Results**   Table 1 provides a performance comparison of EAD against several state-of-the-art diffusion-based methods. From our experiments, we draw three primary conclusions. First, EAD demonstrates superior performance across all evaluated metrics, with its generated molecules closely approximating the distribution of the true dataset. Second, the model exhibits excellent stability, as indicated by the consistently low variance observed across repeated experiments. Third, and most notably, EAD achieves a significant improvement in molecular stability (8.3%) and validity (3.2%) when compared to its base model, EDM, despite being trained with an identical configuration. We note that EAD's sampling time is marginally longer than EDM's due to the additional denoising iterations, with all sampling performed on an Nvidia H800. This increase in computational cost is justified by the substantial improvement in generation quality.

Table 2: Comparison of generation performance on GEOM-DRUG dataset.

| Metrics | EDM | GDM | EDM-Bridge | GeoLDM | UniGEM | EAD (Ours) |
|---|---|---|---|---|---|---|
| Atom sta(%) | 81.3 | 77.7 | 82.4 | 84.4 | 85.1 | **86.3** |
| Valid(%) | 92.6 | 91.8 | 92.8 | **99.3** | 98.4 | 99.1 |

## 5.2   RESULTS ON GEOM-DRUG

**Dataset and Configurations**   We further evaluate EAD's ability to generate large molecules using the GEOM-Drug dataset (Axelrod & Gomez-Bombarelli, 2022) (GEOM). This dataset comprises 430,000 molecules with up to 181 atoms (averaging 44.4 atoms per molecule). Each molecule has multiple conformers and associated energies. Following the configuration in (Hoogeboom et al., 2022), we retain the 30 lowest-energy conformers for each molecule. The models learn to generate the 3D positions and atom types. Additional configuration details are in Appendix D.1.

**Results**   The GEOM dataset is a more challenging benchmark compared to QM9, as its larger molecules and non-equilibrium conformations introduce complexities for bond type prediction and validity assessment based on interatomic distances. Consistent with prior work (Hoogeboom et al., 2022; Feng et al., 2025), our evaluation prioritizes molecular stability and validity. As depicted in Table 2, EAD achieves state-of-the-art performance in molecular stability. For the validity metric, it is important to note that RDKit's aggregation of valid molecular fragments can lead to inflated scores across methods. Despite this, EAD secured the second-highest score, with a minimal difference of 0.2% from the leading method and 0.9% from the maximum possible value (100%).

## 5.3   CONDITIONAL GENERATION

**Configurations**   In this section, we verify the effectiveness of EAD in conditional generation. We introduce molecular properties as conditions into EAD using classifier-free guidance (Ho & Salimans, 2022). To ensure a fair comparison, the experimental configuration and the predictive models for evaluation are consistent with those in (Hoogeboom et al., 2022). The QM9 dataset is partitioned into two sets: one for training the EGNN-based property prediction models, $\varphi(\cdot)$, and the other for training the EAD generation model. Finally, we use $\varphi(\cdot)$ to evaluate 10,000 molecules generated by EAD.

**Results**   Table 3 presents the results of our conditional generation experiments. The "#Atoms" baseline represents a simple model that predicts molecular properties using only the number of atoms. The upper bound on performance, "QM9 (L-bound)", indicates the direct prediction using $\varphi(\cdot)$ on the training data used for generation. We observed that EAD achieved the leading performance across all conditions. Notably, EAD's average improvement is greater than 30% compared to the base model EDM.

Table 3: Mean Absolute Error for molecular property prediction by a EGNN classifier $\phi_c$ on a QM9 subset.

| Task | $\alpha$ | $\Delta\varepsilon$ | $\varepsilon_{\text{HOMO}}$ | $\varepsilon_{\text{LUMO}}$ | $\mu$ | $C_v$ |
|------|----------|---------------------|------------------------------|------------------------------|-------|-------|
| Units | Bohr$^3$ | meV | meV | meV | D | $\frac{\text{cal}}{\text{mol}}$K |
| #Atoms | 3.86 | 866 | 426 | 813 | 1.053 | 1.971 |
| EDM | 2.76 | 655 | 356 | 584 | 1.111 | 1.101 |
| GeoLDM | 2.37 | 587 | 340 | 522 | 1.108 | 1.025 |
| EAD | **2.24** | **564** | **318** | **516** | **0.999** | **0.945** |
| QM9 (L-bound) | 0.10 | 64 | 39 | 36 | 0.043 | 0.040 |

Table 4: Comparison of different sampling strategies.

| Sampling strategy | Atom sta(%) | Mol sta(%) | Valid(%) | V*U(%) |
|-------------------|-------------|------------|----------|--------|
| Synchronous schedule | 98.7 | 81.9 | 91.9 | 90.2 |
| Manual asynchronous schedule | 90.5 | 64.0 | 87.2 | 81.7 |
| Adaptive asynchronous schedule | 99.0 | 90.3 | 95.1 | 93.2 |

## 5.4 ABLATION STUDY

This section presents ablation studies to investigate key characteristics of EAD, including the universality of its training paradigm and the impact of sampling hyperparameters.

**Training Paradigm** Following the training strategy outlined in Section 3.2, we generated 10,000 molecules using synchronous, manually asynchronous, and adaptive asynchronous schedules. The manual timestep schedule employed a step-like noise pattern from (Chen et al., 2024), with details provided in Appendix D.2. From Table 4, we draw two conclusions. First, the synchronous schedule yields results comparable to EDM, confirming that our training paradigm is general enough to encompass synchronous diffusion. Second, the manual asynchronous schedule performs worse, demonstrating that the implicit relationships within a molecular hierarchy cannot be effectively captured by a simple, predetermined schedule. This ablation confirms the universality of our training paradigm while highlighting the need for a carefully designed asynchronous strategy.

**Sampling Hyperparameters** EAD's performance is governed by two key hyperparameters: the asynchronous ratio $\lambda$ and the history window $w$. Results in Table 5 show that a large asynchronous ratio can negatively impact performance, indicating that a relatively good initial state, achieved through sufficient synchronous denoising steps, is crucial before applying asynchronous modifications. Furthermore, a larger history window can impair the dynamic schedule's judgment, leading to a decline in performance.

Table 5: Comparison of different hyper-parameter configuration. Gray grid denotes the default configuration.

| Configuration | Mol sta(%) | Valid(%) | Configuration | Mol sta(%) | Valid(%) |
|---------------|------------|----------|---------------|------------|----------|
| $\lambda = 0.8, w = 2$ | 90.3 | 95.1 | $\lambda = 0.9, w = 2$ | 89.1 | 94.0 |
| $\lambda = 0.8, w = 1$ | 90.0 | 94.8 | $\lambda = 0.6, w = 2$ | 64.7 | 79.5 |
| $\lambda = 0.8, w = 5$ | 80.4 | 90.5 | $\lambda = 0.4, w = 2$ | 12.5 | 36.4 |
| $\lambda = 0.8, w = 10$ | 60.2 | 77.8 | $\lambda = 0.2, w = 2$ | 0.5 | 20.8 |

## 6 CONCLUSION

We present EAD, a novel model for 3D molecular generation that employs an asynchronous denoising paradigm. Our experimental results demonstrate that EAD achieves state-of-the-art performance, offering a promising direction for future research in molecular diffusion. A limitation of our approach is its sensitivity to the asynchronous ratio hyperparameter $\lambda$. Future work will focus on developing more robust and adaptive denoising strategies to overcome this limitation.

# REPRODUCIBILITY

Our datasets are all based on open-source datasets.The experimental methodology, data proportions, and hyperparameter settings are detailed in Appendix.Our code and data are included in the anonymous repositories, which we mentioned in abstract.

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

APPENDIX

## A  THE USE OF LARGE LANGUAGE MODELS

In our work, we exclusively use LLMs for writing refinement, which means we first write a piece of text ourselves, and then use the LLM to correct grammar, formatting, and other issues. For our experiments, we only use LLM to fix bug.

## B  SUPPLEMENTARY PRELIMINARIES

### B.1  DETAILS OF 3D MOLECULAR DIFFUSION

#### B.1.1  NOISE SCHEDULE

In this work, we follow the noise schedule in EDM (Hoogeboom et al., 2022). A diffusion process requires defining $\alpha_t$ and $\sigma_t$ for $t = 0, \ldots, T$. Given the relationship $\alpha_t = \sqrt{1 - \sigma_t^2}$, it is sufficient to specify $\alpha_t$. This parameter should decrease monotonically, starting near $\alpha_0 \approx 1$ and ending at $\alpha_T \approx 0$. In this work, we define $\alpha_t$ as:

$$\alpha_t = (1 - 2s) \cdot f(t) + s, \quad \text{where } f(t) = 1 - (t/T)^2,$$

with a precision value of $10^{-5}$ to prevent numerical instability. This schedule resembles the cosine noise schedule proposed in (Nichol & Dhariwal, 2021), but our formulation is notationally simpler. To further ensure numerical stability during sampling, we adopt the clipping procedure from (Nichol & Dhariwal, 2021). Specifically, we compute $\alpha_{t|t-1} = \alpha_t / \alpha_{t-1}$, with $\alpha_{-1} = 1$. The values $\alpha_{t|t-1}^2$ are clipped from below at $0.001$, ensuring that $1/\alpha_{t|t-1}$ remains bounded during sampling. The $\alpha_t$ values are then reconstructed using the cumulative product $\alpha_t = \prod_{\tau=0}^{t} \alpha_{\tau|\tau-1}$.

The signal-to-noise ratio (SNR) is defined as $\text{SNR}(t) = \alpha_t^2 / \sigma_t^2$. Following (Kingma et al., 2021), we introduce the negative log-SNR curve $\gamma(t) = -(\log \alpha_t^2 - \log \sigma_t^2)$, where $\sigma_t^2 = 1 - \alpha_t^2$. This function, $\gamma(t)$, is monotonically increasing and allows for precise computation of all necessary components. For example:

- $\alpha_t^2 = \text{sigmoid}(-\gamma(t))$,
- $\sigma_t^2 = \text{sigmoid}(\gamma(t))$,
- $\text{SNR}(t) = \exp(-\gamma(t))$.

### B.1.2 Optimization Objective

Recall that the distribution of the diffusion reverse process is represented as:

$$p(\boldsymbol{z}_s|\boldsymbol{z}_t) = \mathcal{N}(\boldsymbol{z}_s|\boldsymbol{\mu}_{t \to s}(\hat{\boldsymbol{x}}, \boldsymbol{z}_t), \sigma_{t \to s}^2 \mathbf{I}). \tag{14}$$

With the choice $s = t - 1$, a variational lower bound on the log-likelihood of $\boldsymbol{x}$ given the generative model is given by:

$$\log p(x) \geq \mathcal{L}_0 + \mathcal{L}_{\text{base}} + \sum_{t=1}^{T} \mathcal{L}_t, \tag{15}$$

where $\mathcal{L}_0 = \log p(\boldsymbol{x}|\boldsymbol{z}_0)$ models the likelihood of the data given $\boldsymbol{z}_0$, $\mathcal{L}_{\text{base}} = -\text{KL}(q(\boldsymbol{z}_T|\boldsymbol{x})|p(\boldsymbol{z}_T))$ models the distance between a standard normal distribution and the final latent variable $q(\boldsymbol{z}_T|\boldsymbol{x})$, and

$$\mathcal{L}_t = -\text{KL}(q(\boldsymbol{z}_s|\boldsymbol{x}, \boldsymbol{z}_t)|p(\boldsymbol{z}_s|\boldsymbol{z}_t)) \quad \text{for } t = 1, \ldots, T.$$

While in this formulation the neural network directly predicts $\hat{\boldsymbol{x}}$, (Ho et al., 2020) found that optimization is easier when predicting the Gaussian noise instead. Intuitively, the network is trying to predict which part of the observation $\boldsymbol{z}_t$ is noise originating from the diffusion process, and which part corresponds to the underlying data point $\boldsymbol{x}$. Specifically, if $\boldsymbol{z}_t = \alpha_t \boldsymbol{x} + \sigma_t$, then the neural network $\phi$ outputs $\hat{} = \phi(\boldsymbol{z}_t, t)$, so that:

$$\hat{\boldsymbol{x}} = (1/\alpha_t)\, \boldsymbol{z}_t - (\sigma_t/\alpha_t)\hat{} \tag{16}$$

As shown in (Kingma et al., 2021), with this parametrization $\mathcal{L}_t$ simplifies to:

$$\mathcal{L}_t = \mathbb{E}_{\sim \mathcal{N}(\mathbf{0}, \mathbf{I})}\left[\frac{1}{2}(1 - \text{SNR}(t-1)/\text{SNR}(t))\| - \hat{}\|^2\right] \tag{17}$$

In practice the term $\mathcal{L}_{\text{base}}$ is close to zero when the noising schedule is defined in such a way that $\alpha_T \approx 0$. Furthermore, if $\alpha_0 \approx 1$ *and* $\boldsymbol{x}$ is discrete, then $\mathcal{L}_0$ is close to zero as well.

### B.2 SO(3) Equivariance

Given any transformation parameter $g \in G$, a function $\varphi : \mathcal{X} \to \mathcal{Y}$ is called equivariant to $g$ if it satisfies:

$$T'(g)[\varphi(x)] = \varphi(T(g)[x]), \tag{18}$$

where $T'(g) : \mathcal{Y} \to \mathcal{Y}$ and $T(g) : \mathcal{X} \to \mathcal{X}$ denote the corresponding transformations over $\mathcal{Y}$ and $\mathcal{X}$, respectively. Invariance is a special case of equivariance where $T'(g)$ is an identity transformation. In this paper, we mainly focus on the $SO(3)$ equivariance and invariance, since it is closely related to the inductive bias of molecules[1]. In other words. The backbone and prediction head should adhere to equation 18.

### B.3 Equivariant Model

EDM utilizes a lightweight neural network known as **E(n) Equivariant Graph Neural Networks (EGNNs)** (Satorras et al., 2021b), and we adopt this approach in our work. EGNNs are a specialized type of Graph Neural Network designed to satisfy the equivariance constraint.

In our framework, we model interactions among all atoms by constructing a fully connected graph $\mathcal{G}$, where each node $v_i \in \mathcal{V}$ represents an atom. Each node $v_i$ is associated with 3D coordinates $\boldsymbol{x}_i \in \mathbb{R}^3$ and feature vectors $\boldsymbol{h}_i \in \mathbb{R}^d$.

The core of EGNNs lies in the **Equivariant Convolutional Layers (EGCL)**, which iteratively update node coordinates and features. Specifically, the $(l+1)$-th layer updates are computed as:

$$\mathbf{m}_{ij} = \phi_e\left(\boldsymbol{h}_i^l, \boldsymbol{h}_j^l, d_{ij}^2, a_{ij}\right), \quad \boldsymbol{h}_i^{l+1} = \phi_h\left(\boldsymbol{h}_i^l, \sum_{j \neq i} \tilde{e}_{ij}\mathbf{m}_{ij}\right),$$

$$\boldsymbol{x}_i^{l+1} = \boldsymbol{x}_i^l + \sum_{j \neq i} \frac{\boldsymbol{x}_i^l - \boldsymbol{x}_j^l}{d_{ij} + 1} \phi_x\left(\boldsymbol{h}_i^l, \boldsymbol{h}_j^l, d_{ij}^2, a_{ij}\right), \tag{19}$$

---

[1]Invariance of translation is trivially satisfied by taking the relative positions as inputs.

where $l$ denotes the layer index, $d_{ij} = \|\boldsymbol{x}_i^l - \boldsymbol{x}_j^l\|_2$ is the Euclidean distance between nodes $v_i$ and $v_j$, and $a_{ij}$ represents optional edge attributes. To enhance numerical stability, the coordinate update in equation 19 normalizes the difference $(\boldsymbol{x}_i^l - \boldsymbol{x}_j^l)$ by $d_{ij} + 1$, following the approach in (Satorras et al., 2021a). Additionally, an attention mechanism infers soft edge weights $\tilde{e}_{ij} = \phi_{inf}(\mathbf{m}_{ij})$.

All learnable components—$\phi_e$, $\phi_h$, $\phi_x$, and $\phi_{inf}$—are implemented as fully connected neural networks. The overall EGNN architecture comprises $L$ EGCL layers, which collectively apply a non-linear transformation to the input coordinates and features: $\hat{\boldsymbol{x}}, \hat{\boldsymbol{h}} = \text{EGNN}[\boldsymbol{x}^0, \boldsymbol{h}^0]$. This transformation inherently satisfies the equivariance property.

## C  MODEL DETAILS

### C.1  MOLECULAR SCAFFOLD

A molecular scaffold is a core structural framework of a molecule that defines its essential topology and connectivity, while abstracting away specific functional groups or substituents. Scaffolds are widely used in cheminformatics and drug discovery to classify and analyze molecules based on their structural similarities. By focusing on the scaffold, researchers can identify common structural motifs across diverse compounds, which is particularly useful for understanding structure-activity relationships (SAR) and designing new molecules with desired properties.

There are several types of scaffolds, including:

- Murcko Scaffolds: Introduced by Bemis and Murcko, this scaffold is derived by removing all side chains and retaining only the ring systems and linkers between them.
- Framework Scaffolds: Similar to Murcko scaffolds but may include additional structural features like non-ring linkers.
- Generic Scaffolds: Further abstract the structure by replacing specific atoms or bonds with generic placeholders.

**Implementation** RDKit is a powerful open-source cheminformatics toolkit that provides tools for working with molecular structures, including scaffold extraction. We can use the Murcko scaffolds using RDKit:

```
from rdkit.Chem import Scaffolds
scaffold = Scaffolds.MurckoScaffold.GetScaffoldForMol(mol)
```

### C.2  CONDITIONAL GENERATION

EAD's ability to generate molecules of variable size significantly enhances its potential for conditional generation. Given a target molecular property $y$, the denoising distribution can be written as $\mathbf{z}_s \sim p_\theta(\mathbf{z}_s|\mathbf{z}_t, y)$. Some chemical properties, such as absolute energy, are closely tied to the size of the molecule. EAD can automatically determine the optimal size that aligns with the desired property. Moreover, by capturing the relationship between molecular size and properties, EAD is capable of generating molecules with properties that fall outside the distribution observed during training. In contrast, methods like EDM (Hoogeboom et al., 2022), which perform denoising based on randomly sampled sizes, may fail to discover the optimal molecular structure. Furthermore, the dual sampling of both size and properties in EDM introduces additional computational overhead.

## D  DETAILS OF EXPERIMENTS AND SUPPLEMENTARY EXPERIMENTS

### D.1  IMPLEMENTATION DETAILS

**QM9.** On QM9, the EAD is trained using a causal EGNN with 256 hidden features and 9 layers. The models are trained for 3000 epochs with a batch size of 1024. The models are saved every 50 epochs when the validation loss is lower than the previously obtained number. The diffusion process uses $T = 1000$. Training takes approximately 2 days on two NVIDIA H800 GPUs.

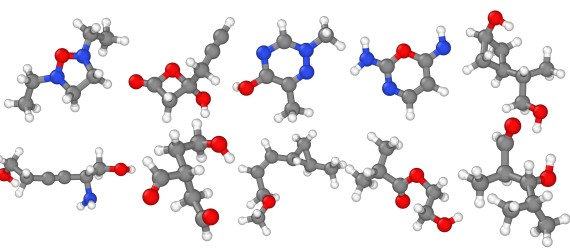

Figure 2: Extra samples generated by EAD trained on the QM9 dataset.

**GEOM.** On GEOM, the EAD is trained using a causal EGNN with $256$ hidden features and $4$ layers. The models are trained for $13$ epochs, which is around $1.2$ million iterations with a batch size of $64$. Training takes approximately $5$ days on four NVIDIA H800 GPUs.

## D.2 MANUAL TIMESTEP SCHEDULE

In this section, we provide a handcrafted asynchronous schedule, which is used in our ablation study. This schedule originates from asynchronous denoising in the video domain (Chen et al., 2024), where videos have explicit causal chains. Following the pattern of videos, this schedule applies higher noise levels to the later frames in the sequence. This scheduling matrix defined as:

$$\mathcal{K} = \begin{bmatrix} T & \dots & T & | & T & \dots & T & |\dots| & T & \dots & T \\ T-1 & \dots & T-1 & | & T & \dots & T & |\dots| & T & \dots & T \\ T-2 & \dots & T-2 & | & T-1 & \dots & T-1 & |\dots| & T & \dots & T \\ & \ddots & & | & & \ddots & & | & | & & \ddots & \\ 1 & \dots & 1 & | & 2 & \dots & 2 & |\dots| & M & \dots & M \\ 0 & \dots & 0 & | & 1 & \dots & 1 & |\dots| & M-1 & \dots & M-1 \\ & \ddots & & | & & \ddots & & | & | & & \ddots & \\ 0 & \dots & 0 & | & 0 & \dots & 0 & |\dots| & 1 & \dots & 1 \\ 0 & \dots & 0 & | & 0 & \dots & 0 & |\dots| & 0 & \dots & 0 \end{bmatrix}. \tag{20}$$

In this matrix, the column axis corresponds to decreasing noise levels, while the row axis represents the sequential generation of the atomic structure. We introduce a new window size $u$, within which atoms share the same noise level. The boundaries of each window are denoted by dashed lines in equation 20. The denoising process progresses row-by-row, iterating left-to-right across columns according to the noise levels specified by $\mathcal{K}$. In the final row, the noise levels of all atoms are reduced to $0$, converging to clean data. In our ablation study, the window size $u$ is set to $1$.

## D.3 SUPPLEMENTARY EXPERIMENTS

**Generation for Unseen or Rare Molecular Sizes.** During the generation process, the EDM model exhibits a notable limitation: when processing a graph of size $n$, it primarily generates molecular features corresponding to molecules of the same size $n$ present in its training set. To investigate this size bias, we conducted an experiment: we challenged EDM to generate 10,000 molecules of unseen or rare sizes. For comparison, our method (EAD) does not require predefined molecular sizes for sampling. Thus, we randomly extracted 5,000 molecules of comparable sizes from EAD's overall generated results. Upon evaluating the uniqueness of the generated molecules (see Table 6), we observed that EDM frequently generates identical molecules. We attribute EAD's superior performance to its inherent ability to incorporate and generate molecular features of varying sizes during the generation process, leading to greater diversity.

Table 6: Uniqueness and novelty over 10000 molecules with defined molecular size.

| Molecular Size | EDM Uniqueness (%) ↑ | EAD Uniqueness (%) ↑ |
|---|---|---|
| 30 (0 molecules in the training set) | 35.68 | 44.33 |
| 29 (25 molecules) | 45.09 | 60.18 |
| 28 (0 molecules) | 45.61 | 65.40 |

