# OpenReview forum: "Equivariant Asynchronous Diffusion: An Adaptive Denoising Schedule for Accelerated Molecular Conformation Generation"
_ICLR.cc/2026/Conference — ICLR 2026 Conference Withdrawn Submission_

### Official Review · Reviewer_mMfW · 2025-10-30

**Soundness:** 3
**Presentation:** 2
**Contribution:** 2
**Rating:** 4
**Confidence:** 3

**Summary:**

Synchronous diffusion models struggle to capture the inherent causal relationships within the molecular hierarchical structure.  To address these limitations, this paper proposes the Equivariant Asynchronous Diffusion (EAD) model, which combines the advantages of both types of models. The EAD model incorporates a stable asynchronous noise training paradigm and a dynamic sampling denoising strategy to generate more reasonable and stable molecules. Experiments show that EAD achieves improvements on the QM9 and GEOM-Drug molecule generation benchmark.

**Strengths:**

1. The idea of combining autoregressive (AR) modeling with diffusion is interesting, offering a novel perspective.

2. The experimental results demonstrate decent performance, validating the practical potential of the proposed method.

**Weaknesses:**

1. Failure to address the core challenge of asynchronous training: The paper acknowledges that fully asynchronous diffusion models suffer from combinatorial explosion. To mitigate this, the authors restrict training to a "narrow asynchronous interval"—implying the maximum timestep difference between different dimensions is limited to 2C. This creates a trade-off: A large C exacerbates training difficulty. A small C makes the method barely distinguishable from fully synchronous training, diminishing the value of the "asynchronous" design.

2. Lack of experimental justification for the need for synchronization: The sampling strategy is intuitive: it retains high noise for less-converged molecular dimensions to align denoising progress across all dimensions. However, from a research logic standpoint, the authors should add an experiment to quantify the misalignment of denoising progress across dimensions in the original synchronous generative model. This would directly justify the necessity of the proposed synchronization mechanism.

**Questions:**

1. What value of C was used in the experiments? Could you provide an ablation study to demonstrate how varying C impacts model performance (e.g., generation quality, training efficiency) ?

2. Why is the sampling time longer than that of EDM? Could you specify the actual diffusion schedule used (e.g., SNR decay rate, timestep count)? Additionally, please provide case studies to verify if the method can to some extent "establish a hierarchical structure within molecules (e.g., preferential emergence of molecular scaffolds)"—as claimed in the introduction for AR methods.

Some further discussions:

3. Given the combinatorial explosion issue mentioned in Weakness 1, could it be possible to propose priors to constrain the timestep combinations across dimensions and reduce complexity? For example: Leveraging chemical domain knowledge (e.g., training heavy atoms first, then hydrogen atoms) to prioritize dimension-wise timestep order. Making the timestep intervals between dimensions learnable (rather than fixed by C) to adapt to data characteristics?

4. While EDM is a classic baseline, its performance is relatively low compared to recent advances. Several studies have improved EDM and achieved better results (e.g., GeoBFN, UniGEM(BFN), Straight-line diffusion model(SLDM)). Is there potential to achieve state-of-the-art (SOTA) performance by combining your framework with any advanced diffusion model?

If main issues are well addressed, I would consider raising my evaluation.

---

### Official Review · Reviewer_TxNc · 2025-10-30

**Soundness:** 2
**Presentation:** 2
**Contribution:** 2
**Rating:** 4
**Confidence:** 4

**Summary:**

This paper proposes the Equivariant Asynchronous Diffusion (EAD) model for 3D molecular conformation generation, aiming to overcome the limitations of autoregressive and synchronous diffusion models. EAD introduces an asynchronous denoising schedule that captures hierarchical dependencies among atoms while maintaining molecule-level coherence. In addition, a dynamic mechanism is employed to adaptively determine denoising timesteps for each atom.

**Strengths:**

1. The asynchronous denoising schedule allows each atom to be generated at its own timestep, effectively capturing the hierarchical structure between core and peripheral atoms — a conceptually interesting design.

2. The model design supports variable-length molecular generation, which introduces a degree of novelty.

3. The authors have released their code, enhancing reproducibility and transparency.

**Weaknesses:**

1. Although the model assigns asynchronous timesteps to simulate molecular hierarchy, it remains unclear whether such hierarchical dynamics are truly learned in practice — for instance, whether core atoms converge faster while peripheral atoms converge more slowly. The paper would benefit from an empirical or visual analysis of this behavior.

2. Since each atom has its own timestep, it is important to clarify how this affects generation efficiency compared to baseline models.

3. The conditional generation experiments are somewhat limited; key baselines such as GeoBFN and UniGEM are missing from the comparison table.

**Questions:**

See weakness

---

### Official Review · Reviewer_LnWp · 2025-10-31

**Soundness:** 2
**Presentation:** 3
**Contribution:** 2
**Rating:** 2
**Confidence:** 4

**Summary:**

The paper proposes Equivariant Asynchronous Diffusion (EAD) for 3D molecular generation. This method assigns per-atom timesteps during training via a constrained independent sampling window $[-C, C]$ around a global baseline $t*$. During sampling, it uses a two-stage schedule: an initial synchronous phase followed by an adaptive per-atom advancement based on a velocity proxy $h_i^* = \|z_{k-1,i} - z_{k,i}\|_2$. The method also proposes a mechanism for variable-size generation by including dummy atoms with biased noise. On QM9, EAD improves over its base model (EDM) on stability and validity metrics. On GEOM-Drugs it reports decent atom stability. Ablations vary the asynchronous ratio $\lambda$, history window $w$, and schedule type.

**Strengths:**

* The paper explores applying asynchronous diffusion to 3D molecular generation, extending prior work in other domains (e.g., text) to the SE(3)-equivariant setting.
* The proposed training method, using a constrained window $t^* \pm C$, is a simple and computationally feasible way to train a model that can handle asynchronous states.

**Weaknesses:**

* The novelty of the core idea is limited. Asynchronous diffusion scheduling has been explored in other domains (e.g., text, video), and this work is a direct application of that concept to molecules.
* The motivation of interpolating between auto-regressive (AR) and standard diffusion to capture molecular hierarchy seems to be an overclaim. The training window $C$ is very small ($C \approx 9$ for QM9) compared to the total timesteps ($T=1000$). It is unclear how such a narrow, local-in-time window can act as a meaningful "interpolation" between AR and diffusion, or capture global hierarchical properties like scaffolds vs. functional groups.
* The method is extremely sensitive to its key hyperparameters. Table 5 shows that straying slightly from the ($\lambda=0.8, w=2$) configuration causes performance to collapse (e.g., $\lambda=0.6$ drops stability from 90.3% to 64.7%; $w=5$ drops it to 80.4%). This suggests the gains may come from a bag of tricks specific to QM9 rather than a generalizable principle. Needs experiment on other dataset as well and report the hyperparameters on there as well to confirm the robustness.
* Key design choices lack justification or ablation studies.
    * The choice of $C \approx M_{max}$ (max atoms) is arbitrary and lacks theoretical or empirical support.
    * The hyperparameter $C$ is not studied in an ablation.
    * The value of $C$ used for the large-molecule GEOM-Drug dataset (where $M_{max}=181$) is not specified. If $C \approx 181$, it contradicts the $C \ll T$ setup from QM9. Also, it is unclear about the sensitivity of the choice of $C$ for different datasets.
* The experimental evaluation is insufficient.
    * On GEOM-Drug (Table 2), only stability and validity are reported. Key metrics like uniqueness, novelty, and computational cost (runtime/NFEs) are missing.
    * The "accelerated" claim in the title is unsubstantiated. The paper shows similar runtime on QM9. (0.20 sec/sample for EDM 0.21 sec/sample for EAD which shows it's even slower than the baseline) and provides no runtime analysis for GEOM-Drug.
    * The performance gain over other SOTA methods (e.g., UniGEM) is marginal (Table 1). Also, there exist several work that surpass the results of the method proposed by the authors. Needs more extensive comparison with the SOTA baselines.

**Questions:**

* Could the authors provide a stronger justification for the link between the time window $C$ and the max atom count $M_{max}$? What $C$ value was used for the GEOM-Drug experiments, and why? An ablation study on $C$ would be valuable.
* The adaptive scheduler is extremely sensitive to $\lambda$ and $w$ (Table 5). Does this not suggest the method is a brittle heuristic rather than a robust adaptive principle? How can users be expected to tune this for new datasets?
* Why was the $L_2$ velocity (Eq. 12) chosen as the criterion for "stalling" an atom? Were any chemically-motivated metrics (e.g., based on valence, interatomic forces, or energy) considered?
* For the GEOM-Drug results, could the authors please provide the missing metrics: uniqueness, novelty, and runtime (sec/sample)? Without these, it is difficult to assess the method's utility on larger molecules.
* The motivation is to capture hierarchy. However, $C \ll T$ implies the model only ever sees states that are "almost" synchronous. How can this local-in-time asynchrony lead to learning global molecular hierarchies?

I'm willing to raise my score if all concerns in the weaknesses and questions are properly addressed.

---

### Official Review · Reviewer_KfAz · 2025-11-01

**Soundness:** 2
**Presentation:** 3
**Contribution:** 2
**Rating:** 4
**Confidence:** 2

**Summary:**

This paper proposes Equivariant Asynchronous Diffusion (EAD) for 3D molecular generation, which assigns different noise levels to different atoms during diffusion instead of denoising all atoms simultaneously. The method uses a velocity-based metric to adaptively determine denoising priority, aiming to capture molecular hierarchical structures. Experiments on QM9 and GEOM-Drug datasets show improvements in molecular stability and validity over baseline EDM.

**Strengths:**

1. The motivation to combine autoregressive and diffusion paradigms for molecular generation is sound and addresses real limitations of existing methods. The asynchronous denoising concept is intuitive—molecules indeed have hierarchical structures where certain components (scaffolds) should reasonably be prioritized over others (functional groups).
2. The authors provide a valuable theoretical contribution by proving that synchronous diffusion (EDM) is a special case of their framework, and demonstrate reliable performance gains over EDM across multiple metrics without requiring additional training epochs.
3. A notable advantage is that a single trained EAD model can be deployed with different sampling strategies—synchronous, manual asynchronous, or adaptive asynchronous—without retraining.

**Weaknesses:**

1. The paper's central claim is that asynchronous denoising can capture molecular hierarchy, but the paper provides no theoretical analysis or empirical evidence for why this method should correlate with structural importance. For example, there is no visualization or analysis showing which atoms actually get denoised first in practice—do carbon scaffolds genuinely get prioritized? Do hydrogen atoms get denoised later?
2. Table 5 reveals a catastrophic weakness in the method's robustness. The authors acknowledge this sensitivity as a limitation but provide essentially no principled approach for setting $\lambda$. The window size $w$ is also critical but under-explored.
3. The paper lacks a thorough computational cost analysis. The sampling overhead is underreported: while the increase from 0.20 to 0.21 seconds seems modest, how many extra denoising iterations are actually performed? For adaptive scheduling, this could vary significantly per molecule, yet no statistics are provided. And there is result analysis on training cost.
4. The improvements over recent strong baselines are marginal.

**Questions:**

1. What does "accelerated" in the title refer to? I This needs to be clearly addressed or the title should be changed. I don't seem to see any evidence of 'accelerated' reflected in the paper.
2. Can you provide concrete evidence (visualization, quantitative metrics, chemical analysis) that the model actually learns to prioritize structurally important atoms?
3. Can you add analysis on hyperparameters and methods of determining $lambda$ and $w$ for new datasets without extensive grid search?
4. Can you provide a thorough computational cost analysis?

---

### Note · Authors · 2026-01-21

**Comment:**

We sincerely thank all the reviewers for their valuable feedback. While our work introduces an interesting idea for molecular generation, we acknowledge that there are still some issues that need to be addressed. To this end, we have decided to withdraw our submission and continue to improve our work. We hope to present a more refined version in the future.

**Withdrawal Confirmation:**

I have read and agree with the venue's withdrawal policy on behalf of myself and my co-authors.